# Cryopreservation and the Freeze–Thaw Stress Response in Yeast

**DOI:** 10.3390/genes11080835

**Published:** 2020-07-22

**Authors:** Elizabeth Cabrera, Laylah C. Welch, Meaghan R. Robinson, Candyce M. Sturgeon, Mackenzie M. Crow, Verónica A. Segarra

**Affiliations:** Department of Biology, High Point University, High Point, NC 27268, USA; lcabrera@highpoint.edu (E.C.); lwelch@highpoint.edu (L.C.W.); mrobins4@highpoint.edu (M.R.R.); csturgeo@highpoint.edu (C.M.S.); kcrow@highpoint.edu (M.M.C.)

**Keywords:** yeast, cryopreservation, cryoprotectants, freeze–thaw stress response

## Abstract

The ability of yeast to survive freezing and thawing is most frequently considered in the context of cryopreservation, a practical step in both industrial and research applications of these organisms. However, it also relates to an evolved ability to withstand freeze–thaw stress that is integrated with a larger network of survival responses. These responses vary between different strains and species of yeast according to the environments to which they are adapted, and the basis of this adaptation appears to be both conditioned and genetic in origin. This review article briefly touches upon common yeast cryopreservation methods and describes in detail what is known about the biochemical and genetic determinants of cell viability following freeze–thaw stress. While we focus on the budding yeast *Saccharomyces cerevisiae*, in which the freeze–thaw stress response is best understood, we also highlight the emerging diversity of yeast freeze–thaw responses as a manifestation of biodiversity among these organisms.

## 1. Introduction

Yeasts comprise a diverse group of unicellular fungi. Collectively, these organisms have served as a workhorse for both industrial and research applications, in part due to their genetic and biochemical malleability. Industrial innovations and scientific progress in the research laboratory have been greatly facilitated by the ability to curate and build large collections of these organisms by cryopreserving (preserving by freezing) yeast for long periods of time, if not indefinitely. In this review, we summarize what is known about the methods and cellular processes associated with yeast cryopreservation, with the majority of this knowledge being especially applicable to the budding yeast *Saccharomyces cerevisiae*. This species is of interest due to the availability of detailed studies characterizing its freeze–thaw stress response, a cellular process by which cells recover from cryopreservation and resume growth.

We also explore the connection between the freeze–thaw stress response and other stress responses in *S. cerevisiae*. The freeze–thaw stress response is important in part because of its integration with a larger network of stress responses, including cold and heat shock and oxidative stress. Some of the genes and pathways that confer freeze–thaw resistance are linked to survival of additional forms of stress, including nutrient sensing signaling, the cell wall integrity (CWI) pathway, and proteasomal function. Other resistance factors can be broadly categorized as biochemical protectants, including lipid components that confer membrane flexibility and metabolites that prevent ice crystal formation and allow for reactive oxygen species (ROS) scavenging. As research moves beyond *S. cerevisiae* alone, comparisons to cold-adapted species have yielded additional insights into a spectrum of biodiversity in yeast freeze–thaw stress responses.

## 2. Reagents and Methods for Yeast Cryopreservation

### 2.1. Overview of Cryopreservation

There are several methods that can be used to cryopreserve yeast [1,2,3]. Cryopreservation of yeast is usually achieved by growing the strain of interest in rich growth medium before adding a cryoprotective agent to cells prior to freezing. In general, yeast strains can be preserved and stored indefinitely in 25% (v/v) glycerol at temperatures of −60 °C or lower [4]. Because of their ease of implementation, methods utilizing glycerol as a cryoprotectant and freezing temperatures from −60 to −80 °C are the most commonly used. We briefly describe these protocols below and their reported effects. We then discuss their relationship to the freeze–thaw stress response.

### 2.2. Cryopreserving Yeast Grown on Plates

The yeast strain of interest is grown on the surface of a rich medium plate, collected by scraping with a sterile applicator and suspended in 25% (v/v) sterile glycerol (in water) solution inside of a 2 mL screw cap cryovial [1,2,3]. Screw cap cryovials are preferred over snap top tubes as the latter can pop open unexpectedly at low temperatures. After tightening of the screw cap and mixing to homogeneity, the cryovial is placed at −60 °C or lower [4]. Instead of rich medium plates, selective or drop-out medium plates may be used.

### 2.3. Cryopreserving Yeast Grown in Medium Broth

Yeast of interest is grown to late log phase or saturation in liquid rich medium and mixed with an equal volume of sterile glycerol at a concentration anywhere between 20% and 50% (v/v in water) inside a 2 mL screw cap cryovial [1,2,3]. Alternatively, the desired culture of cells can be mixed with sterile rich medium containing the desired concentration (v/v) of glycerol. After tightening of the screw cap and mixing to homogeneity, the cryovial is placed at −60 °C or lower [4]. Instead of rich medium broth, selective or drop-out medium broth may be used.

### 2.4. Reviving Yeast Cells after Freezing

Cryopreserved yeast can be thawed out when desired and allowed to revive and resume growth. This can be achieved by transferring a small amount of the frozen sample onto the desired growth medium, whether on plates or in liquid culture [4]. When collecting a small amount of the frozen sample, care is generally taken not to completely thaw out the entire glycerol stock, as viability of the cells remaining in the frozen stock will decrease with each freeze/thaw cycle. To minimize chances of the stock thawing out completely, placing the stock vial on dry ice while collecting cells to revive is sometimes recommended. For most applications, successful cell revival following cryopreservation is essentially evaluated qualitatively by the formation of colonies on plates or growth in the desired liquid medium of interest.

## 3. Freeze–Thaw Stress Response in Yeast

### 3.1. Freezing and Thawing as a Form of Stress and a Cause of Damage

Freezing and thawing are forms of stress that can cause physiological injury to the cell. Cellular damage from the freezing step of the process is predominantly due to the formation of ice crystals and cellular dehydration. The extent of this damage ultimately depends on the rate at which a sample is frozen and whether intracellular or extracellular ice crystals predominate [1,2,3,5,6]. At slow rates of freezing, extracellular forms of ice crystals are thought to form quickly and be in the majority, driving intracellular water transport outwards and ultimately leading to cellular damage from intracellular dehydration (Figure 1) [7]. On the other hand, at rapid rates of freezing, intracellular freezing occurs more quickly and intracellular ice crystals are responsible for most of the cellular damage (Figure 1) [8]. Slower freezing (temperature decreasing less than 7 °C/min) has been found to yield higher cell viabilities than fast freezing [6,9]. Rate of freezing can also be influenced by strain- or species-specific characteristics that include cell shape and surface area-to-volume ratio. While most of the damage in cells exposed to freeze–thaw stress results from the freezing step, damage arises through the thawing process as well, in part due to enhanced formation of ROS. Interestingly, survival rates following freeze–thaw stress are not dependent on thawing temperature [9].

### 3.2. Molecules That Minimize Freeze–Thaw Stress Damage during Cryopreservation

While recovery from freeze–thaw stress is a necessary element of cryopreserving a yeast strain with the goal of culturing it later, the presence of protective molecules can limit the extent of this damage. To minimize the effects of low temperatures on cells as described above, prior to cryopreservation, a cryoprotectant such as glycerol or dimethyl sulfoxide (DMSO) is added to cells [1,2,3]. Both glycerol and DMSO are capable of diffusion across cellular membranes, so that they are considered permeable or permeating cryoprotectants. Cryoprotective molecules work to protect cells from damage by reducing the crystallization of water molecules. In fact, without addition of a cryoprotective agent, yeast cell viability is lost as an exponential function of freezing duration [9]. Non-permeating molecules such as trehalose and sorbitol cannot diffuse across biological membranes but can be used as cryoprotectants [2,5,10,11]. More specifically, the non-permeating cryoprotectant sorbitol has been found to preserve competent intact yeast cells for efficient electroporation [10].

### 3.3. Investigation of Freeze–Thaw Stress Induction and Tolerance

While cryopreserving and reviving yeast is sufficient to activate the freeze–thaw stress response, research laboratories studying sensitivity/tolerance have interrogated this response in a more systematic and controlled manner. In general, freeze–thaw stress is induced in the research laboratory by harvesting cells at the desired growth phase and washing/resuspending them in 50–100 mM sodium or potassium phosphate buffer (pH 7.0) to an A_600_ of ~3.0 before placing them at −20 °C for 2 h [9,12]. Cells grown and harvested in rich medium can also directly be exposed to freeze–thaw stress without washing/resuspending in additional buffers [13]. Rates of freezing and thawing can be monitored using microprocessor-based thermometers placed inside the tube containing the cells of interest [9]. To measure the cell viability or survival after freeze–thaw insults, a colony formation assay is often used [9,12,13]. In this assay, cells are diluted in nutrient-rich growth medium so that the suspension, when plated onto nutrient-rich plates and grown at 30 °C for a couple of days, will yield an expected amount of single colonies that can be easily counted (~100–200 cells or colonies) [14]. The resulting colonies are counted, compared to the unstressed control, and considered a measure of cell viability or survival [9,12,13,14].

## 4. Cellular Factors That Can Influence Freeze–Thaw Stress Tolerance of *S. cerevisiae*

### 4.1. Overview

The freezing and thawing tolerance of budding yeast is influenced by factors such as growth phase, mitochondrial function, membrane composition, metabolites such as trehalose and *alpha*-ketoglutarate, proteins such as aquaporins and N-acetyltransferase, multistress tolerance genes, target of rapamycin (TOR), Ras/cyclic adenosine monophosphate (cAMP), and CWI signaling pathways, as well as the proteasomal catabolic mechanism (Figure 2). While the field does not yet understand in detail the mechanisms by which these influences are integrated to regulate yeast freeze–thaw tolerance, we discuss below what is known about each factor.

### 4.2. Growth Phase

Yeast cell growth can be separated into lag phase; logarithmic phase, a slower diauxic growth phase during which the predominant method for energy harvesting shifts from glycolysis to aerobic utilization of ethanol; and a stationary or quiescent phase after carbon sources are depleted and growth saturation is reached. Interestingly, wild-type yeast cells display different freeze–thaw stress sensitivities during each of these growth stages. Yeast generally display high freeze–thaw tolerance during stationary phase and low tolerance during early exponential or log phase [9]. Cells in lag phase gradually become more sensitive to freeze–thaw stress as they approach logarithmic growth [9]. While screens have been performed to identify mutants with defective freeze–thaw stress responses [13,15,16], the relationship between this phenotype and genes controlling growth phase has not been extensively investigated.

### 4.3. Mitochondrial Function and Cellular Respiration

Mitochondrial function and the ability to carry out cellular respiration are requirements for the full freeze–thaw stress tolerance in budding yeast [9,17]. A potential explanation for the relationship between mitochondrial function and freeze–thaw stress is the fact that both mitochondrial damage and specimen rehydration/thawing are known sources for ROS generation [9,17].

### 4.4. Membrane Composition

Loss of cell membrane integrity due to physical damage is a key element of freeze–thaw stress, and the flexibility of the cell membrane plays an important role in cell survival [1,2,3,18,19]. Cold temperatures tend to compress membranes, decrease membrane fluidity, and increase membrane susceptibility to rupture, particularly when the proportion of saturated fatty acids is high [18]. Yeast strains with high proportions of lipids that foster membrane fluidity, such as polyunsaturated fatty acids, ergosterol (the predominant sterol found in fungi, equivalent to cholesterol), and phospholipids with linoleyl residues, possess enhanced freeze–thaw resistance [9,18,19].

### 4.5. Trehalose Accumulation

Trehalose, a disaccharide composed of two glucose monomers, is a relatively stable carbohydrate that can act as a protectant for subcellular structures in times of stress [2,5,20,21,22,23]. Trehalose fulfills this role by stabilizing the internal water structure and fortifying the cell membrane against extreme environmental fluctuations [5,22]. Yeast strains with high freeze–thaw tolerance produce large amounts of trehalose during cryopreservation, positively correlating resistance with the sugar’s presence [5,22,23]. Trehalose has also been shown to independently increase stress tolerance for diploid *S. cerevisiae* mutants [22]. The preservative ability of trehalose production has been observed in other yeast strains as well. *Schizosaccharmyces pombe* exhibits enhanced tolerance to both heat shock and freeze–thaw stress when induced to exogenously express the trehalose-6-phosphate synthase gene [23].

### 4.6. Alpha-ketoglutarate

Alpha-ketoglutarate (AKG) is a Krebs cycle intermediate metabolite. It is a “nitrogen scavenger” and a glutamate and glutamine amino acid precursor [12,24]. *S. cerevisiae* grown on an AKG-supplemented medium prior to freezing exhibit enhanced freeze–thaw tolerance [25]. The ability of AKG to improve freeze–thaw tolerance in yeast may be linked to its role in amino acid synthesis and the increased antioxidant capacity observed in AKG-treated cells [25]. Interestingly, incubation of *S. cerevisiae* in 10 mM AKG solution also protected against carbohydrate-induced cell death [12].

### 4.7. N-Acetyltransferase

N-acetyltransferase (NAT) is an enzyme that catalyzes the transfer of acetyl groups from acetyl-coenzyme A to metabolites such as l-azetidine-2-carboxylic acid (AZC) in yeast [26,27]. Mpr1 function allows for reduction of intracellular reactive oxidation levels, protecting yeast cells from increased ROS production and oxidative stress that arise from freeze–thaw cycles [27]. Mpr1 is constitutively expressed in yeast cells, and its expression increases as a result of freeze–thaw stress [26]. Mpr1 is only present in the *S. cerevisiae* laboratory strain Sigma1278b (not S288C) [27].

### 4.8. Aquaporins

The protective effect of aquaporins first emerged through the observation that deletion of the genes encoding them increases sensitivity to freeze–thaw stress and that their overexpression increases cell freeze–thaw tolerance [28,29,30,31,32,33,34]. Aquaporins are a family of water channel proteins that mediate the transport of water across cell membranes in numerous species [28,29,30,31,32,33,34]. While orthodox aquaporins are permeable only to water, aquaglyceroporins are permeable to water, glycerol, and other small uncharged molecules [28,29,30]. The *S. cerevisiae* genome contains two orthodox aquaporins, coded for by the *AQY1* and *AQY2* genes, and two aquaglyceroporins [28,29,30,31,32,33,34]. *AQY1* becomes abundantly expressed when yeast are starved for nutrients. The other, *AQY2*, is expressed by exponentially growing cells. Deletion of any of these genes in laboratory yeast strains sensitizes them to freeze–thaw stress, while their overexpression, as well as heterologous expression of the human aquaporin gene *hAQP1*, improved freeze–thaw tolerance [31]. hAQY1’s ability to improve freeze tolerance of yeast supports the idea that rapid osmotically-driven water efflux from cells during the freezing process lowers the intracellular water content, subsequently reducing intracellular ice crystal formation and damage, resulting in increased freeze–thaw tolerance [31]. Similarly, expression of higher levels of endogenous aquaporins in the plasma membrane are thought to allow for faster water efflux during freeze–thaw stress [31,32,33]. All of these findings support a role for plasma membrane water transport activity in the determination of freeze–thaw tolerance of yeast [31,32,33,34].

### 4.9. Cold-Induced Stress Changes in Gene Expression

Cold or freeze–thaw stress induces a specific pattern of differential gene expression in *S. cerevisiae*, indicating that the cellular response to this stress is regulated at least in part at the transcriptional level [13,15,16,35,36,37,38,39]. This cold-induced pattern is distinct from heat shock-induced changes in gene expression [13,35]. On the other hand, studies have identified a cold shock-activated gene named *TIP1* (temperature-inducible protein) and two homologs, *TIR1* and *TIR2* [35]. Other genes activated by cold stress are *TPI1*, *MMS2*, *PAK1*, *ERG10*, *SEC11*, *SSD1*, *IMH1*, *YNL278w*, and *YFL030w* [13]. *TPI1* and *ERG10* in particular are positively correlated with cryo-resistance in *S. cerevisiae* [13]. In fact, *ERG10* overexpression increases freeze–thaw tolerance [13].

### 4.10. Multistress Tolerance Genes and the Intersection between Freeze–Thaw Stress and Other Cellular Stress Coping Mechanisms

While some of the changes in gene expression induced by exposure to cold or freezing conditions are unique, others are part of a more generalized stress response and can be similarly triggered by other forms of stress [13,15,16,35,36,37,38,39,40,41,42,43,44]. Yeast and other microorganisms have evolved rapid molecular responses that coordinate the repair of stress-induced cellular damage and to build up tolerance against continued exposure to new or repeated forms of stress. These stress-induced genes (i.e., multistress tolerance genes) mediate a cross-protective effect by which exposure to mild stress triggers the development of tolerance not only to higher doses of the same stress, but also to stress caused by other agents. Pre-treatment of cells with hydrogen peroxide, cycloheximide, mild heat shock, or NaCl stress, for example, confers cross-protection against freeze–thaw stress [9]. Hydrogen peroxide has been used in some studies to identify mutants that exhibit tolerance not only to H_2_O_2_ itself but also to ethanol, high glucose concentrations, chronological aging, and freeze–thaw stress [37]. Additionally, while we have been able to isolate these mutant strains with increased tolerance to a multiplicity of stresses, it is only recently that we have started to understand the molecular and functional basis for the increased stress resistance of these mutations [42].

Genes that are upregulated during periods of oxidative stress to increase tolerance to multiple forms of cellular stress are controlled by the stress-responsive transcription factors Msn2 and Msn4. These two transcription factors are regulated through phosphorylation and mediate the effects of multiple signal transduction pathways on cellular stress tolerance. In the absence of stress, Msn2 and Msn4 transcriptional activity is negatively regulated by phosphorylation events that block their nuclear import, including by the cAMP-responsive kinase PKA and the glucose-sensing kinase Snf1 [38]. These findings are consistent with the roles of Msn2 and Msn4 in cell survival of diverse stresses such as oxidative damage and entry into stationary phase, and their function in integrating information from the TOR, Ras/cAMP, glucose sensing, and CWI pathways [39,40,41]. Additional genes that function in generalized stress responses including freeze–thaw stress include *ATH1* and *TPS1*. Cells defective for the enzyme Ath1 accumulate trehalose and exhibit enhanced survival, not only under freezing conditions, but also in the presence of dehydration or ethanol toxicity [43]. Conversely, Tps1 mutants that counteract trehalose synthesis are less tolerant to temperature-related stresses [44].

### 4.11. Ras/cAMP Signaling Pathway

The Ras-cyclic adenosine monophosphate (cAMP) signaling pathway has been found to play an important role in the freeze–thaw stress response [41]. This may relate to its well-characterized function in regulating the entry and survival of cells in the non-dividing, quiescent, stationary phase. Yeast cells enter stationary phase upon nutrient (nitrogen or carbon) starvation, often associated with an increase in freeze–thaw tolerance. In fact, mutations that abrogate the Ras/cAMP pathway function increase sensitivity to freeze–thaw stress [9].

### 4.12. Proteasome

The proteasome is a protein complex that carries out selective degradation of polyubiquitinated proteins. Proteasome function helps control a variety of cellular processes, including recovery from freeze–thaw stress.

The role of the proteasome in freeze–thaw stress is not well-understood, but the transcriptional activation of genes encoding proteasome components is required for freeze–thaw stress survival [45]. The function of the proteasome in this context may be to degrade proteins damaged during the processes of freezing and thawing.

### 4.13. Autophagy

Other forms of cellular stress response have not yet been sufficiently investigated for roles in recovery from freeze–thaw stress in yeast specifically. Autophagy, for example, is a catabolic cellular process that is induced upon cellular damage or nutrient starvation. During autophagy, cells use large double-membraned vesicles to sequester cellular components and deliver them to the vacuole for degradation and recycling. Through this pathway, yeast cells in stationary phase, without readily available nutrients, can recycle unneeded or damaged cellular parts to sustain critical metabolism and strive for survival. While autophagy triggers such as the starvation for either nitrogen or carbon sources has been shown to lead to increased levels of freeze–thaw tolerance [9], the dependence of the freeze–thaw stress response on yeast autophagy has not been directly tested. On the other hand, Antarctic midge insects have been shown to survive extreme dehydration in their natural environment through the upregulation of autophagy genes [46,47]. Other circumstantial evidence that suggest a link between autophagy and freeze–thaw stress includes the observation that the Ras/cAMP pathway regulates both autophagy and freeze–thaw stress [9,41].

## 5. Emerging Lessons from Antarctic Yeasts

### 5.1. Overview

The vast majority of what is known about the freeze–thaw response in yeast has been characterized using laboratory strains of *S. cerevisiae*. An emerging strategy to complement this traditional approach is to expand studies into wild species of yeast in order to better understand adaptations that allow survival in extreme environments [48]. Interestingly, the freeze–thaw stress tolerance of yeasts can vary depending on the species, a manifestation of biodiversity among these organisms. For example, cultivable yeasts have been isolated from Antarctic soil samples. These are mostly reported to grow in temperatures from 4–30 °C and to be basidiomycetus, belonging predominantly to the *Cryptococcus* species [49]. These yeasts have adapted to low-temperature environments that are dry, low in nutrient availability, and have high exposure to solar radiation. While initial characterization of these yeast species has been carried out, little is known about how their cellular processes may have adapted to the cold and how they might be different from mesophilic yeast species such as *S. cerevisiae*. Below, we review what is known about these yeast species and the molecular basis for their high freeze–thaw tolerance.

### 5.2. Stress Resistance and Metabolism

Some of the most notable differences between cold-adapted yeast and *S. cerevisiae* relate to their ability to withstand stress. In general, when compared to *S. cerevisiae*, cold-adapted yeast display increased resistance to stresses such as changes in temperature and salinity [50]. Additionally, while it is unknown to what degree these metabolic characteristics can confer freeze–thaw resistance, Antarctic yeast species have been reported to grow on and assimilate carbon sources including alpha-D-glucose, sucrose, D-trehalose, turanose, D-xylose, dextrin, and salicin [48,49]. Yeasts adapted to cold environments are in fact known to secrete a variety of hydrolytic enzymes, enabling them to use a large number of materials as carbon sources. On average, Antarctic yeasts have been found to secrete 4–7 enzyme activities, often including lipase, alkaline phosphatase, invertase, amylase, cellulose, and pectinase [39,40]. Using [^3^H] leucine incorporation as a measure of metabolic activity, it has been found that both Antarctic and, albeit to a lesser extent, *Saccharomyces cerevisiae* yeasts are both able to carry out metabolism in the −5 °C to −15 °C temperature range [51].

### 5.3. Trehalose Production

Trehalose biosynthesis has been shown to play a key role in enhancing the freeze–thaw stress tolerance of cold-adapted yeast [52]. Genomic and quantitative real-time PCR studies in the Antarctic yeast *Pseudozyma* sp. NJ7 have identified the presence and regulated expression of genes encoding trehalose phosphate synthase (TPS1) and trehalose phosphate phosphatase (TPS2) in times of salinity and temperature stress [52]. Moreover, *Pseudozyma*-synthesized trehalose can be successfully extracted, isolated, and exogenously used as a cryoprotectant for these same cells [52].

### 5.4. Membrane Composition

Antarctic yeasts have been found to produce high proportions of ergosterol, total fatty acids, and linoleic acid (phospholipids with linoleyl residues), suggesting that maintaining membrane fluidity is one of the ways in which they have adapted to the cold [19,48]. One of the obvious benefits of this membrane composition would be increased tolerance to freeze–thaw stress.

### 5.5. Antifreeze Proteins

Antifreeze proteins (AFPs) are a family of large, structurally diverse proteins that are secreted by cold-adapted organisms [48,53]. AFPs have the ability to bind to ice and prevent its crystallization, limiting the growth of ice crystals and reducing cell damage due to freeze–thaw stress [40,41]. Interestingly, the genes encoding these proteins have been uniquely identified in Antarctic species of yeast, and protein-mediated cryoprotection against ice crystal formation may not occur to the same extent in mesophilic species such as *S. cerevisiae* [53].

### 5.6. Complex Adaptation of Antarctic Yeast to Cold Environments

All of the characteristics described above are thought to contribute to cold adaptation in Antarctic yeast, and none of these responses alone are responsible for the high tolerance of these species to freeze–thaw stress [48,53]. This suggests a complex interplay of cellular factors to survive cold environments, similar to that observed in mesophilic yeast.

## 6. Conclusions

Cellular stress responses have evolved in a highly integrated way, such that exposure to one stress can broadly cross-protect cells against diverse future stresses. As a result, much of what is currently known about the basis of freeze–thaw stress tolerance is linked to mechanisms that are shared among a variety of different stress responses. New investigations of cold-adapted yeast species may enable the field to focus more directly on mechanisms designed specifically to confer freeze–thaw stress tolerance. At the same time, new lines of investigation in *S. cerevisiae* may be able to reveal new insights into the molecular and signaling overlap between freeze–thaw stress response and other well-known stress responses such as autophagy and proteasomal degradation. Recent progress made in these adjacent fields represents an opportunity to investigate their potential connection to freeze–thaw stress. The increasingly detailed understanding of cryopreservation and freeze–thaw stress in yeast is highly relevant to both research and industrial applications of these organisms.

## Figures and Tables

**Figure 1 genes-11-00835-f001:**
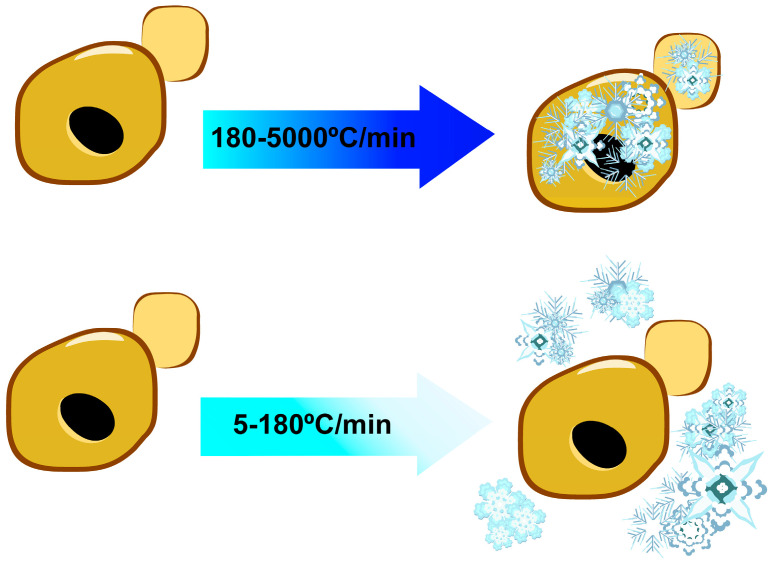
Cellular damage resulting from freezing depends on cooling rate. Slower rates of freezing minimize the generation of intracellular ice crystals, decreasing cell damage and increasing cell viability upon thawing.

**Figure 2 genes-11-00835-f002:**
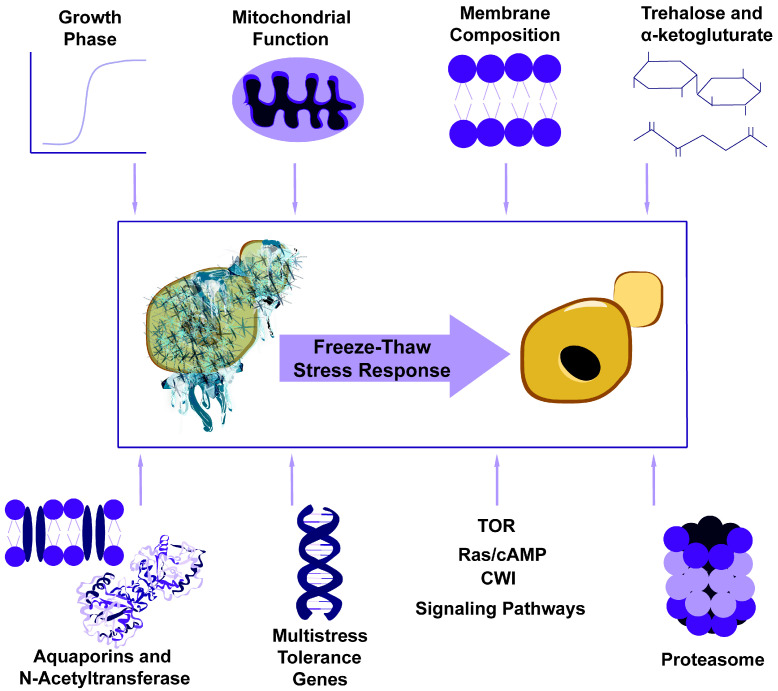
Cellular factors influencing the freeze–thaw stress tolerance of *Saccharomyces cerevisiae*.

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
