# Peer review of "Cryopreservation and the Freeze–Thaw Stress Response in Yeast"

_genes, 2020, doi:10.3390/genes11080835_

Round 1
Reviewer 1 Report
Reviewer’s comments for the authors:
Cryopreservation is a method by which live cells can be frozen/stored (below -60 ºC) for an indefinite period with the possibility to regrow them as and when needed. Yeast Saccharomyces cerevisiae is a powerful tool for industrial innovations and scientific research because of its genetic and biochemical malleability. Because of its ability to withstand the stress that occurred due to cryopreservation (known as freeze-thaw stress), it facilitates building a large collection of genetically engineered yeast strains in the laboratories for a long period. Generally, yeast cells can be stored below -60 ºC just by resuspending (stationary phase) cells in 25% (v/v) glycerol and this is quite a common practice being used by all the yeast labs world-wide. However, the molecular mechanism of how cells respond to/withstand this stress is not clearly understood. Also, there is limited literature on developing new methods for cryopreservation, making cells more competent to withstand the freeze-thaw stress and understanding its molecular mechanism.
In their manuscript, Cabrera et al. provide a comprehensive review of the work published on the cryopreservation methods and the freeze-thaw stress response. They summarize the current state of knowledge about the freeze-thaw stress response in yeast and cellular factors that influence stress tolerance. At the end, they discussed emerging lessons from Antarctic yeasts which are naturally trained to withstand low temperatures and in what aspects they differ from the mesophilic yeasts. I would consider this review as an important compilation for the current state of knowledge for cryopreservation and freeze-thaw stress response. It is a well-structured, well-written article that needs minor corrections and incorporation of a few important references to improve the overall style and content.
Major comments:
- The authors should discuss the cryopreservation of competent yeast cells for efficient electroporation. Ref.: Suga et al. Yeast 2000 (Cryopreservation of competent intact yeast cells for efficient electroporation). Including this information will attract more readers to this article, as many labs use electroporation with yeast cells for DNA manipulations and incorporation of fluorescence dyes.
- Figure 1 legend: Authors should list cellular factors in the text and remove them from the legend as they are self-evidence from the figure, however, these factors are missing from the text (Page 3, lines 112-113)
- Authors may consider including this reference: Jeong et al. 2012 (The viability and recovery of S. cerevisiae after freezing at -84 ºC with different concentrations of glycerol).
- Authors may consider including the graph for the effect of cooling rate on S. cerevisiae viability from Dumont et al. Appl Environ Microbiol 2004 (Cell size and water permeability as determining factors for cell viability after freezing at different cooling rates, Figure 1). It would be also interesting to readers if the concept of surface-to-volume ratio (S/V ratio) and hydraulic permeability (Lp) can be introduced in the context of water crystallization in and out of the yeast cell and heat transfer (from the same paper).
- A simple concept for optimized cryopreservation states that slow freezing (~1 ºC/min) and fast thawing (~200 ºC/min) is ideal. Authors have mentioned a freezing rate of <7 ºC/min, however, they missed to mention the optimized thawing rate. Authors may include this information in section 3.1
Minor comments:
- In the reference list, there is a repetition of the reference numbers. E.g. 1. [1], 2. [2] and so on. Please correct this throughout the reference list.
- Either use freeze-stress or freeze-thaw stress consistently throughout the manuscript.
- Make the font size consistent throughout the manuscript. In particular, I can see larger fonts on page 7, lines 234-237.

Author Response
- Reviewer’s comments 1: “The authors should discuss the cryopreservation of competent yeast cells for efficient electroporation. Ref.: Suga et al. Yeast 2000 (Cryopreservation of competent intact yeast cells for efficient electroporation). Including this information will attract more readers to this article, as many labs use electroporation with yeast cells for DNA manipulations and incorporation of fluorescence dyes.”
- Author’s response 1: Thank you for this recommendation. This citation and reference has been added.
- Reviewer’s comments 2: “Figure 1 legend: Authors should list cellular factors in the text and remove them from the legend as they are self-evidence from the figure, however these factors are missing from the text (Page 3, lines 112-113)”
- Author’s response 2: Thank you for this recommendation. This recommendation was adopted.
- Reviewer’s comments 3: “Authors may consider including this reference: Jeong et al. 2012 (The viability and recovery of cerevisiae after freezing at -84 ºC with different concentrations of glycerol).”
- Author’s response 3: We tried many times to find this peer-reviewed reference and did not find it. Our apologies.
- Reviewer’s comments 4: ”Authors may consider including the graph for the effect of cooling rate on cerevisiae viability from Dumont et al. Appl Environ Microbiol 2004 (Cell size and water permeability as determining factors for cell viability after freezing at different cooling rates, Figure 1). It would be also interesting to readers if the concept of surface-to-volume ratio (S/V ratio) and hydraulic permeability (Lp) can be introduced in the context of water crystallization in and out of the yeast cell and heat transfer (from the same paper).”
- Author’s response 4: Thanks for the recommendation. While we think that this content is beyond the scope of this paper, we did add one figure to the manuscript. This figure (see first figure of paper; there are two now) describes the relationship between cooling rate and the degree of intracellular versus extracellular water crystallization. We think this will be of great interest to the readers. Thanks for the suggestion.
- Reviewer’s comments 5: ”A simple concept for optimized cryopreservation states that slow freezing (~1 ºC/min) and fast thawing (~200 ºC/min) is ideal. Authors have mentioned a freezing rate of <7 ºC/min, however, they missed to mention the optimized thawing rate. Authors may include this information in section 3.1”
- Author’s response 5: Thank you for your suggestion. We tried to find a reference to cite along with the thawing rate provided by reviewer. Despite many tries, we were unable to. Our apologies.
- Reviewer’s comments 6: “In the reference list, there is a repetition of the reference numbers. E.g. 1. [1], 2. [2] and so on. Please correct this throughout the reference list.”
- Author’s response 6: Thank you, this has been corrected.
- Reviewer’s comments 7: Either use freeze-stress or freeze-thaw stress consistently throughout the manuscript.
- Author’s response 7: Thank you, this has been corrected.
- Reviewer’s comments 8: “Make the font size consistent throughout the manuscript. In particular, I can see larger fonts on page 7, lines 234-237.”
- Author’s response 8: Thank you, this has been corrected.
Reviewer 2 Report
The manuscript consist of an interesting review concerning the state-of-the-arts related to the procedures of cryo-preservation of Saccharomyces cerevisiae and other yeast species and about the freeze-thaw stress response in the above microbes. Moreover, the Authors proficiently summarize the cellular factors that are likely to affect the tolerance of S. cerevisiae to the stress induce by the freeze-thaw treatment
The paper is concise, well written and the literature data are clearly reported and discussed. It deals with a topic of relevant interest and supply an exhaustive description of present knowledge about the argument. I think that this is a worthy paper with interesting information to report.
However, the section concerning the Antarctic yeast should be enriched by citing and discussing the following papers:
Yin, H., Wang, Y., He, Y., Xing, L., Zhang, X., Wang, S., ... & Miao, J. (2017). Cloning and expression analysis of tps, and cryopreservation research of trehalose from Antarctic strain Pseudozyma sp. 3 Biotech, 7(5), 343.
Ballester-Tomás, L., Prieto, J. A., Gil, J. V., Baeza, M., & Randez-Gil, F. (2017). The Antarctic yeast Candida sake: Understanding cold metabolism impact on wine. International journal of food microbiology, 245, 59-65.
Amato, P., Doyle, S., & Christner, B. C. (2009). Macromolecular synthesis by yeasts under frozen conditions. Environmental microbiology, 11(3), 589-596.
Author Response
Reviewer’s comments: “However, the section concerning the Antarctic yeast should be enriched by citing and discussing the following papers:
Yin, H., Wang, Y., He, Y., Xing, L., Zhang, X., Wang, S., ... & Miao, J. (2017). Cloning and expression analysis of tps, and cryopreservation research of trehalose from Antarctic strain Pseudozyma sp. 3 Biotech, 7(5), 343.
Ballester-Tomás, L., Prieto, J. A., Gil, J. V., Baeza, M., & Randez-Gil, F. (2017). The Antarctic yeast Candida sake: Understanding cold metabolism impact on wine. International journal of food microbiology, 245, 59-65.
Amato, P., Doyle, S., & Christner, B. C. (2009). Macromolecular synthesis by yeasts under frozen conditions. Environmental microbiology, 11(3), 589-596.”
Author’s response: Thank you for your suggestions. We have used the indicated additional references to enrich the section on Antarctic yeast. Thanks.
Reviewer 3 Report
This is review concisely covers the topic of cryopreservation. It highlights key mechanisms that have added a lot in understanding the freeze thaw response in budding yeast.
- Authors can add more to introduction why this freeze thaw response is important.
- They have mentioned about all types of freeze thaw responses, if they could make a table with references and genes. That will give readers a better understanding of the text.
- Any examples of genes which are known to effect by growth phase freeze thaw?
- Is there a reference for this statement “Interestingly, survival rates following freeze-thaw stress are not dependent on thawing temperature” Page 2, line 83
- Authors should add references in places where they have cited specific genes and facts, like page 6, Line 193 and 194.
Author Response
Reviewer’s comment 1: “Authors can add more to introduction why this freeze thaw response is important.”
Author’s response 1: Thank you for your suggestion. We have added an additional paragraph to the introduction.
Reviewer’s comment 2: “They have mentioned about all types of freeze thaw responses, if they could make a table with references and genes. That will give readers a better understanding of the text.”
Author’s response 2: Thank you for your suggestion. To make the manuscript more readable, we have simplified the text so that only key genes are mentioned. We have done this by deleting large laundry list of genes and referring readers to the relevant references. For this reason, an additional table is no longer needed. Thank you.
Reviewer’s comment 3: “Any examples of genes which are known to effect by growth phase freeze thaw?”
Author’s response 3: Thank you for your suggestion. We do not know of any such genes, we have indicated this in the manuscript.
Reviewer’s comment 4: “Is there a reference for this statement “Interestingly, survival rates following freeze-thaw stress are not dependent on thawing temperature” Page 2, line 83”
Author’s response 4: Thank you for your suggestion. A reference has been added where requested.
Reviewer’s comment 5: “Authors should add references in places where they have cited specific genes and facts, like page 6, Line 193 and 194.”
Author’s response 5: Thank you for your suggestion. References have been added where requested.